# A Web Survey to Evaluate the Thermal Stress Associated with Personal Protective Equipment among Healthcare Workers during the COVID-19 Pandemic in Italy [note 1]

**DOI:** 10.3390/ijerph18083861

**Published:** 2021-04-07

**Authors:** Alessandro Messeri, Michela Bonafede, Emma Pietrafesa, Iole Pinto, Francesca de’Donato, Alfonso Crisci, Jason Kai Wei Lee, Alessandro Marinaccio, Miriam Levi, Marco Morabito

**Affiliations:** 1Institute of Bioeconomy, National Research Council (IBE-CNR), 50019 Florence, Italy; alfcrisci@gmail.com (A.C.); marco.morabito@ibe.cnr.it (M.M.); 2Centre of Bioclimatology, University of Florence (UNIFI), 50144 Florence, Italy; 3Occupational and Environmental Medicine, Epidemiology and Hygiene Department, Italian Workers’ Compensation Authority (INAIL), 00143 Rome, Italy; m.bonafede@inail.it (M.B.); e.pietrafesa@inail.it (E.P.); a.marinaccio@inail.it (A.M.); 4Physical Agents Sector, Regional Public Health Laboratory, 53100 Siena, Italy; iole.pinto@uslsudest.toscana.it; 5Department of Epidemiology Lazio Regional Health Service, ASL ROMA 1, 00147 Rome, Italy; f.dedonato@deplazio.it; 6Human Potential Translational Research Programme, Yong Loo Lin School of Medicine, National University of Singapore, Singapore 117593, Singapore; phsjlkw@nus.edu.sg; 7Department of Physiology, Yong Loo Lin School of Medicine, National University of Singapore, Singapore S117593, Singapore; 8Global Asia Institute, National University of Singapore, Singapore S119076, Singapore; 9N.1 Institute for Health, National University of Singapore, Singapore S117456, Singapore; 10Institute for Digital Medicine, National University of Singapore, Singapore S117456, Singapore; 11Singapore Institute for Clinical Sciences, Agency for Science, Technology and Research (A*STAR), Singapore 117609, Singapore; 12Epidemiology Unit, Department of Prevention, Central Tuscany Local Health Authority, 50135 Florence, Italy; miriam.levi@uslcentro.toscana.it

**Keywords:** occupational safety and health, adaptation strategy, PPE, global warming, heat stress

## Abstract

The pandemic has been afflicting the planet for over a year and from the occupational point of view, healthcare workers have recorded a substantial increase in working hours. The use of personal protective equipment (PPE), necessary to keep safe from COVID-19 increases the chances of overheating, especially during the summer seasons which, due to climate change, are becoming increasingly warm and prolonged. A web survey was carried out in Italy within the WORKLIMATE project during the summer and early autumn 2020. Analysis of variance (ANOVA) was used to evaluate differences between groups. 191 questionnaires were collected (hospital doctor 38.2%, nurses 33.5%, other healthcare professionals 28.3%). The impact of PPE on the thermal stress perception declared by the interviewees was very high on the body areas directly covered by these devices (78% of workers). Workers who used masks for more than 4 h per day perceived PPE as more uncomfortable (*p* < 0.001) compared to the others and reported a greater productivity loss (*p* < 0.001). Furthermore, the study highlighted a high perception of thermal stress among healthcare workers that worn COVID-19-PPE and this enhances the need for appropriate heat health warning systems and response measures addressed to the occupational sector.

## 1. Introduction

In 2020, humanity faced, and is still facing, the most severe pandemic the world has been confronted with since the pandemic of “Spanish flu” back in 1918. Between the 1 January 2020 to the 8 February 2021, 105.805.951 confirmed cases of COVID-19 and 2.312.278 deaths worldwide have been reported by the World Health Organization (https://covid19.who.int/, accessed on 5 March 2021). The most affected continent were the Americas [1] with 47,122,757 confirmed cases, followed by Europe with 35,620,266 confirmed cases up to the 8 February 2021. The Covid-19 pandemic presents a massive unplanned experiment [2] and with regards to the occupational setting, in particular healthcare workers (HCW), have experienced a substantial increase in working hours (increase in shifts). This category of workers has been exposed to an increased risk of SARS-CoV-2 infection due to their frequent exposure to infected individuals, but at the same time also to psychological distress, fatigue, occupational stigma, depression and anxiety [3,4]. In addition, during the warm season, these symptoms can be exacerbated by heat stress imposed on the body for the enhanced use of Personal Protective Equipment (PPE), which is necessary to reduce the risk of disease transmission [5,6]. As also stated by the WHO [7], PPE wear by HCW varies a lot according to the work environment, the type of job activity and the type of patient (patients with confirmed coronavirus disease or not). Considering these aspects, PPE wear by HCW habitually includes face masks with filters (N95 respirator), face shields, goggles and closed work shoes [8].

However, the use of PPE, while necessary to keep workers safe from COVID-19, also increases the chances of overheating [9] and consequently amplify the risk of heat stress for these workers, [10,11]. The human body produces heat that increases according to the physical effort and therefore significantly increases in case of intense work activities [12]. The body must dissipate this excess of heat to the environment through sweat evaporation, convection and conduction [6]. The outgoing removal of metabolic body heat is limited by COVID-19′s PPE which, compared with standard medical scrubs, has approximately double the evaporative resistance [13]. Furthermore, this resistance can increase over 10-fold with added layers and with full encapsulation of the head and neck [14]. Consequently, limited heat loss combined with potentially high sweat rates, thermal discomfort, and fatigue can occur rapidly [15] leading to critical health conditions such as dehydration and hyperthermia. In addition, COVID-19′s PPE worn for long shifts and associated with environmental heat may further aggravates effects such as skin reactions [16], respiratory difficulties, nausea, digestive discomfort, headaches [17] and mental health impacts [18,19]. Furthermore, as the COVID emergency has made necessary to call back retired medical staff to work, these are at greater risk of COVID-19 health complications as well as heat stress due to their age [20,21,22].

In this situation, it is crucial to have a better understanding of the environmental working conditions and thermal stress perceived by HCW. In a context where the priority is the prevention of SARS-CoV-2 infection, it seems to be very important, to develop strategies to mitigate the effects of heat conditions, including for example monitoring of local thermal stress in the work place and the development of a specific heat-health warning system for occupational sectors [9,23,24,25]. These potential adverse physical and mental effects, experienced by frontline HCW, may further impact the already struggling healthcare system during the pandemic [6]. A few studies have assessed heat stress due to PPE in the healthcare sector during the COVID-19 pandemic in international settings through surveys [3,26]. In the frame-work various ad hoc questionnaires have been developed to evaluate the perceived level of heat stress experienced by healthcare professionals and how this situation has influenced their physical, cognitive and emotional sphere in working life [6,27,28].

In Italy, the “WORKLIMATE” project (“Impact of environmental thermal stress on workers’ health and productivity: intervention strategies and development of an integrated heat and epidemiological warning system for various occupational sectors”) started in June of 2020 (project details are available at https://www.worklimate.it, accessed on 6 April 2021). The aim of the project is to improve the knowledge basis and awareness on health effects of environmental thermal stress conditions (in particular heat) on workers. As part of the project activities, a web survey was carried out to investigate the impact of COVID-19′s PPE among healthcare workers. The e-research is, in fact, a new investigative tool, widely used in Countries with high internet usage. According to the literature [29,30], the advantages of e-research over a traditional study (telephone, post or personal interview) are: (a) speed of detection (the online survey times are certainly lower than research carried out in a traditional way); (b) monitoring and real-time analysis of the data (following the insertion/recording of the data, a summary and immediate analysis of the trend is possible); (c) cost- effectiveness (internet interviews are cheaper than similar surveys conducted using traditional methods); (d) reduction of intrusiveness of detection (an online questionnaire is a tool to which the user has decided to answer behind the prompt of very few external agents; this improves the fidelity and spontaneity of the answers); (e) achievement of specific targets favoring the communicative specificity of the survey; (f) use of multimedia (sound, pictures and movies).

The first aim of this study was to assess the impact of COVID-19′s PPE on the environmental thermal stress of HCW engaged in different activities. In addition, information regarding types of PPE, the potential productivity loss and adaptive behaviors carried out to reduce heat stress during the work shift, were also collected. This information could be particularly useful when defining prevention measures in response to heat stress among HCW and to improve their productivity during emergency situations like the COVID-19 pandemic, or other similar future-emergency measures, requiring the same approach as a priority.

## 2. Materials and Methods

A self-administered web-based questionnaire was developed (Appendix A), complemented by an informed consent form, and the participation was voluntary and anonymous. The estimated time to complete the questionnaire was around 15/20 min. Data were collected, stored and analyzed according to the Regulation on the protection of natural persons with regard to the processing of personal data (EU Regulation 2016/679—General Data Protection Regulation—GDPR—application from 25 May 2018).

This activity received the ethical clearance from the Commission for Ethics and Integrity of Research of the National Research Council (CNR) (N. 0009389/2020, 2 June 2020).

### 2.1. Survey Development

The survey (Annex 1) was an adapted version of a tool developed by Lee et al. [6], used in a previous study to assess the knowledge, attitudes, and practices of HCWs in India and Singapore concerning PPE’ usage and heat stress during treatment and care activities. The WORKLIMATE questionnaire was created and administered entirely in Italian language (https://forms.gle/rBbJixexAaBD6m3h9) and consisted of different sections including:-demographic data and characteristics of the worker (question from 1 to 8)-relevant work information (9–13)-heat-exposure-related questions and information about PPE’ usage at work (14–20) -worker’s adaptation to heat stress and behavioral with PPE (21-27)-worker’s knowledge about thermal stress and attitudes towards the PPE’s use (28–46) 

A 5-point Likert scale (1 for strongly disagree and 5 for strongly agree) was used for questions from 28 to 46 concerning the worker’s knowledge about thermal stress and attitudes towards the PPE’s use.

### 2.2. Survey Administration

The questionnaire was prepared using the Google Form online platform (https://www.google.it/intl/it/forms/about/, accessed on 6 April 2021) and was disseminated through the official website and social accounts of the WORKLIMATE project (https://www.worklimate.it; https://www.facebook.com/Worklimate; https://twitter.com/worklimate) as well as through the involvement of Technicians for prevention in the environment and in the workplace. In addition, ad hoc emails were sent to professional associations and advertisements via personal networks and social media accounts of management committee members. 

The questionnaire was administered only to HCW who work in Italian hospitals with a specific focus on Covid departments. The survey was accessible for 5 months, starting at the 1 June and ending at the 31 October 2020.

### 2.3. Study Area and Climatic Characteristics

The study analyzed data of 191 questionnaires collected during the summer and early autumn 2020, in months characterized by temperatures that were, in most of the Italian regions, slightly above the average compared to the climatology 1981–2010, especially in Central and Southern Italy (Figure 1). Between July and August, the thermal anomaly was close to 1.5 °C in some southern regions. We can therefore state that the questionnaire administration period coincided with a warmer summer than the reference climatology.

### 2.4. Data Analysis

The data collected were analyzed using descriptive statistics (i.e., frequency, mean, standard deviation) and analytical tests. The analysis of variance (ANOVA) was used to evaluate differences between groups. The homogeneity of variance was verified with the Levene test. The Brown–Forsythe and Welch tests were used when the homogeneity of variance assumption did not hold for the data. A Principal Component analysis (PCA) with Varimax rotation was carried out and the Cronbach’s Alpha calculation allowed an empirical assessment of the reliability to assess the dimensionality of section “worker’s knowledge on thermal stress and attitudes towards the use of PPE. The results were considered significant at a *p*-value less than 0.05. All analyses were performed using SPSS v25.0 for Windows (IBM, Armonk, NY, USA).

## 3. Results

### 3.1. Descriptive Analysis

191 HCW participated in the self-administered web survey, most of whom (56%) carried out their work activities in South and Central Italy. The sex distribution was homogeneous for the health sector with 132 women (69.1%) and 59 men (30.9%). The average age of participants was 43.7 years (SD ± 11.1), the average height and weight were respectively 169 cm (±8.4) and 69 kg (±14.5). As for body mass index (BMI), 65% of the interviewees fell into the normal weight (BMI < 25) category, while 35% were overweight (BMI > 25). The analyzed sample included many types of professions involved in the healthcare sector with the most HCW represented by hospital doctors (38.2%) and nurses (33.5%). 

Less than 13% of HCW reported they avoid eating on fast days for personal reasons. More than half the responders (about 58%) declared they were involved in activities requiring a high or very high physical effort (Table 1).

About 60% of HCW declared they perceived heat discomfort (from slightly to very hot), despite the prevalent working environment being indoor and air-conditioned (79.1%). Less than 20% perceived slightly or very cold conditions. As expected among HCW, the number of days per week that PPE were used is very high (5.2; ±1) with a claimed average time to put on these garments about 7.1 min (±5.5) at the start of each work shift. Surgical mask were the most used PPE: it was worn for over 4 h a day by 74.3% of workers. N95 mask or FFP2 mask were also widely used and were worn for over 4 h per day by 53.4% of workers. The FFP3 mask was rarely used and it was worn at least 1 h a day by only 15% of the subjects. Gloves were also widely used, 34.6% said they used gloves from 1 to 3 h a day, 28.8% from 4 to 6 h and 15.6% over 6 h. About 32% of worker’s stated that they used 2 pairs of gloves at the same time for 1 to 3 h a day and 25.1% after 4 h. 37.5% of workers wore disposable gowns from 1 to 3 h a day, 26.2% used them for a period between 4 and 6 h, 11% even more than 6 h. Normal gowns were slightly less used and overall only 47.8% said they used it for at least 1 h a day. Even less used were aprons: 9.0% of the participants used them for at least 1 h a day and only 2.1% used them over 6 h. As for eye protection, 59.7% of the participants used them and among workers and about 38.2% wore disposable glasses over 4 h a day (12% over 6 h). Disposable visors were also widely used by healthcare personnel: about 29% said they used them between 1 and 3 h a day, 20% between 4 and 6 h and about 10% over 6 h. Disposable headgear was widely used: 47.2% used it for at least 4 h a day and 64.9% for at least 1 h. As for the foot protection, the most used PPE were the sanitary clogs: over 37% of respondents said they used them for at least 6 h a day, 22% from 4 to 6 h. 46.6% of workers said they used shoe covers too, 16.8% between 4 and 6 h, 6.8% after 6 h. Finally only 55.5% of workers declared there was a company procedure that allowed them to dress and remove PPE during work breaks. 

The impact of PPE on the thermal stress perception declared by the interviewees was very high on the body areas directly covered by the devices (78% of workers). In general, 99% of the participants declared a “hot” heat stress perception during work activity and slightly more than 50% even a “very hot” thermal sensation. The body parts affected by the HCW heat stress perception are depicted in Figure 2.

The lower face part was the body area for which the greatest number of HCW (35.6%) declared very hot sensation: 34% hot and 11% slightly hot; but 13.6% of participants perceived cold. Regard to the hands (27.2%), the armpits (30.4%) and the chest (28.8%), the HCW declared a very hot sensations too and respectively 24.1%, 26.7% and 22.5% hot sensations. According to the interviewees, the upper face part was also affected by hot conditions, in particular 27.2% of the respondents felt very hot, 32.5% hot and 12.6% slightly hot conditions. Less heat stress was perceived on the neck and legs, in fact only 19% and 15.7% declared very hot conditions respectively. 

The symptoms related to heat stress prevalently described were: thirst (58%), excessive sweating (70.7%), general discomfort (51.8%), fatigue (46.1%) and headache (42.9%). Skin reactions (26.7%) and difficulty concentrating (29.3%) were reported too. Many HCW reported adopting strategies to reduce the effects of heat, particularly by often drinking water (56.5%), taking breaks whenever possible (42.4%), wearing light clothing (47.1%), preferring ventilated and cool environments if present (33.5%). Less represented were breathing techniques and only 1 subject declared drinking ice cold drinks. A great number of HCW (81%) self-reported a productivity loss related to heat stress exposure.

### 3.2. Principal Component Analysis

From the Principal Components analysis (PCA) have carried out on “Worker’s knowledge about thermal stress and attitudes towards PPE use” to verify the existence of common dimensions. Three factors that explain 67.1% of the variance emerged from the analysis (Table 2).

The first factor (α = 0.90), which explains the 34.9% of the variance, has been called “Perception of heat stress conditions in the workplace and productivity “because it brings together all the items concerning the subjective impacts of heat stress and the perception of loss of productivity of the worker.

The second factor (α = 0.82), which explains the 16.7% of the variance, has been called “HCW behavior during the working days “because it brings together all the items concerning actual behaviours during work days, what healthcare professionals avoid doing or what is uncomfortable for them to do.

The third factor (α = 0.75), which explains the 15.4% of the variance, has been called “Awareness of good practices” because it brings together all the items concerning some good practices for managing heat stress.

In the factorial solution the items 37, 38, 39, 40, 45, 46 have been excluded.

### 3.3. Differences between HCW Groups

The analysis of variance highlights significant differences between the average scores assigned to different items for different groups. The groups were chosen considering all the aspects that can play a key role in the different thermal perception in the occupational field: geographical area in which the workplace was located; thermal environment exposure; physical and personal characteristics of the worker (gender, age, BMI); kind of work, work effort, PPE characteristic and use (type of PPE, duration of use); worker behavior; company procedures and symptoms. 

For most items, the analysis of variance did not show significant differences between workplaces in different geographical areas (North compared with Central-South Italy) and did not show any significant difference between working environments with or without air conditioning. We also carried out an ANOVA between workers who declared to work in a hot environment (about 60%) and those working in a cold or neutral environment (about 40%) but no significant differences emerged except for the item “My work productivity is reduced when I wear PPE” (*p* < 0.05). In this case, the subjects who worked in a warm environment declared to be more agreement with this item (M = 3.3, SD = 0.1) than who worked in a cold or neutral environment (M = 2.9, SD = 0.1).

The age and gender of the workers did not seem to influence significantly the answers provided by the interviewees too. On the other hand, several physical characteristics, and especially BMI, reveled a significant heat impact (*p* < 0.01) on the reasoning skills of workers. A difference (*p* < 0.05) emerged between the group of overweight or obese subjects (BMI > 25) (M = 4.5, SD = 0.7) compared to normal or underweight workers (BMI < 25) (M = 4.1, SD = 1.0) concerning the effect of heat stress on the impair reasoning.

Moreover, a difference (*p* < 0.05) emerged between hospital doctors and nurses concerning the role of a good hydration and adequate rest between shifts to improve tolerance to heat. In particular, doctors seemed to be more in agree with these two items and declared PPE’s more uncomfortable (M = 4.1, SD = 1.0) compared to what reported by the nurses too (M = 3.7, SD = 1.2).

The groups were chosen considering all the aspects that can play a key role in the different thermal perception in the occupational field. The results of the analysis between groups divided into 4 fundamental issues are shown below: perception of heat stress conditions in the workplace (items 28, 29, 31, 32, 33); perception of productivity loss and PPE use (items 37, 39, 46); behavior during the working days (items 38, 41, 42, 43, 44) and awareness of good practices should be adopted before and during the shift (items 34, 35, 36, 40, 45).

Concerning the first two issues aimed at assessing the perception of heat stress conditions in the workplace and the perception of productivity loss by the worker, a significance emerged from the interviews between different group, linked to the kind of work, the use of glasses, visor and headgear, as well as the thermal sensation related to the use of PPE (Table 3 and Table 4).

Workers who used the headgear for more than 4 h a day (Table 3) and who worked in the company without a specific procedure regarding use of PPE (Table 4), declared a significant (*p* < 0.05) reasoning impairment (items 29). Furthermore, a company procedure to dress PPE was significantly correlated (*p* < 0.01) with a psychological distress associated with heat stress (item 32) and with the awareness that this condition can also affect the commitment at work (item 33). In addition, subjects who reported a very hot thermal sensation and who used the headgear for more than 4 h a day declared a significant (*p* < 0.01) effect of heat stress on their physical health too (item 30).

General practitioners and hospital doctors (M = 4.1, SD = 1.0) considered PPE more uncomfortable (*p* < 0.001) than other healthcare workers (M = 3.4, SD = 1.3) (Table 4). Furthermore, the productivity loss (item 28) was found to be significantly correlated (*p* <0.001) to the perception of thermal sensation due to the use of PPE. Workers who reported a very hot thermal sensation were more aware of the role of PPE in hindering sweat evaporation (item 46) as well as those who used glasses or visors for more than 4 h a day (Table 3). As for the perception of productivity loss, it appeared significantly greater in subjects who declared a very hot thermal sensation (*p* < 0.05), in those who wore more headgear (*p* < 0.05) and highly correlated (*p* < 0.001) with the lack of company procedures to dress PPE (Table 4). Thermal perception and company procedures on the correct use of PPE also played a key role in attributing the productivity loss to the PPE use (item 39). This was confirmed by the fact that workers who declared a different thermal perception between different body parts covered by the PPE were more in agreement with this item. 

On the other hand, as regards the items relating to the issue “HCW behavior during the working days” (items 38, 41, 42, 43, 44) and “awareness of good practices should be adopted before and during the shift” (items 34, 35, 36, 40, 45), the different kind of work, the use of glasses, visor and headgear as well as the thermal sensation related to the use of PPE, showed a significant difference between different groups. (Table 5 and Table 6).

Workers who used glasses, visors and headgear more, also declared a greater difficulty in taking breaks to rehydrate (item 41). This behavior was confirmed by the fact that these workers were (*p* < 0.01 and *p* < 0.001 respectively) agree with the item 42 (I avoid taking breaks to not remove and put on the PPE again) (Table 5). Item 41 was related to the work effort, to the presence of company rest areas and above all to the different thermal perception in the body parts covered by the PPE too (Table 6). Moreover, the workers who revealed a greater work effort reported no rest areas available in the company and declared a great different perception between the body parts covered and uncovered by the PPE. In addition, these HCW also declared difficulties (*p* < 0.05) in taking breaks because too busy (item 38). Finally, the kind of work and the use of the headgear influenced the responses to the item “I avoid taking breaks to not remove and put on again PPE” (43, Table 5).

Interesting results also emerged linked to the items relating to worker awareness of some good practices addressed to increase heat tolerance. For example, the awareness that slush drinks improve the tolerance to heat (item 45) was significantly higher (*p* < 0.001) in subjects who did not use or use little headgear (Table 5) and in companies in which no specific procedures to dress PPE exist (Table 6). The awareness that taking breaks increases heat tolerance (item 34) is correlated to the thermal sensation (Table 5) and to the presence in the company of a specific procedure to dress PPE (Table 6). Furthermore, the kind of work and the work effort (*p* < 0.05) influenced the worker’s awareness that adequate rest between shifts increases heat tolerance (item 36). The behavior adopted by worker before the shift, and in particular the maintenance of a good hydration (item 35) was also considered very important especially by workers who declared a neutral (M = 4.6, SD = 0.7) or slightly warm (M = 3.9, SD = 1.2) thermal sensation, compared to those who said of perceiving very hot (M = 3.7, SD = 1.1) (Table 5).

### 3.4. Masks, Gloves and Other PPE

As highlighted in the descriptive analysis, masks represented one of the most used PPE by HCW and for this reason, their impact on thermal stress perception was thoroughly evaluated taking into account the number of hours and the type of mask used. Many items and therefore many answers provided by HCW were significantly influenced by this equipment. In particular, the awareness that good behavioral practices outside the workplace, such as keeping fit and maintaining a good level of hydration before starting, were significantly (respectively *p* < 0.05, *p* < 0.001) influenced by the use of masks (Figure 3).

HCW who used different types of masks for a total time exceeding 4 h per day (M = 4.2, SD = 1.2) significantly (*p* < 0.001) considered PPE more uncomfortable than those who only used one type of mask for less than 4 h (M = 2.0, SD = 1.4). A very similar result was obtained with the productivity loss perception caused by the use of PPE which was significantly higher (*p* < 0.001) for the first group of HCW (M = 3.7, SD = 1.7) than the second one (M = 1.7, SD = 0.5). Furthermore, HCW who used less masks (fewer types) and for less time revealed a significant (*p* < 0.01) lower awareness of the role that PPE have in hindering the evaporation of sweat. The impact of masks on good practices during work shifts was significant too. In fact, those who used only one type of mask and for less than 4 h, were less motivated to take breaks during the work shift, in this way avoiding to take off and put on PPE (*p* < 0.01), to rehydrate (*p* < 0.05), for drinking and eating (*p* < 0.05), compared to HCW who wore multiple types of masks for more than 4 h a day. 

The use of gloves also had a significant impact on the responses provided by workers (Figure 4). 

In particular, the subjects who used two pairs of overlapping gloves, for at least 4 h a day, perceived a higher productivity loss (M = 3.5, SD = 1.2) and revealed greater difficulty in taking breaks during the work shift to rehydrate too (M = 3.4, SD = 1.4), compared to HCW who used them for short time or who did not use them at all (M = 2.8, SD = 1.2). HCW who worn two pairs of overlapping gloves declared to avoid taking breaks to not remove and put on the PPE again (M = 3.4, SD = 1.4) and even preferred not to eat or drink to avoid going to the toilet (M 3.3, SD 1.4), compared to HWC who did not use gloves or who used them very little (M = 2.4, SD = 1.4).

Other PPE, in particular gown, disposable apron and shoes, seem to less influence the responses, and therefore were associated with a lower perception of the heat-stress-related risk. As for footwear, subjects who wore closed boots, work shoes, shoes cover or sanitary clogs with shoes cover, showed a significant (*p* < 0.05) greater perception (M = 4.3, SD = 1.0) of the role of PPE in hindering the evaporation of sweat, compared to those who did not wear these PPE (M = 3.7, SD = 1.1). In addition, workers that wore closed boots, work shoes or sanitary clogs (M = 3.3, M = 1.5) with shoes cover, avoided drinking and eating with the aim to reduce breaks to use toilet compared to the others (M = 2.5, SD = 1.4).

PPE-use-related symptoms were very common among HCW. With the aim of evaluating their impact on the thermal stress perception, HCW were divided into 6 groups according to the number of symptoms declared in the survey: from group 1 with one symptom to group 6 with more than 6 symptoms. Symptoms significantly influenced the productivity loss perception (F = 4.3, *p* < 0.001) related to the use of PPE (F = 6.2, *p* < 0.001) and to the emotions too (F = 4.1, *p* < 0.001). Furthermore, HCW who declared a higher number of symptoms (more than 4 symptoms), also declared more difficulty in taking breaks because they were too busy (F = 3.5, *p* < 0.001), or because they did not prefer to remove and put on the PPE again (F = 2.9, *p* < 0.01). Finally, they also reported to avoid drinking and eating in order to not to go to the toilet (F = 6.3, *p* < 0.001).

## 4. Discussion

This study represents one of the first surveys to investigate how heat-stress perception among Italian HCW was influenced by the use of COVID-19′s PPE. The knowledge of working conditions and health of workers involved in the healthcare sector, who are currently on the front line of the response to the Covid-19 pandemic, is a priority.

Recently, several studies have demonstrated that age, gender, socioeconomic deprivation, ethnicity could be predictive demographic and social risk factors for COVID-19. Moreover, also hypertension, diabetes and obesity are underlying health conditions that can increase the risk of the disease [31]. The interplay of this underlying conditions and the risk of contracting COVID-19 infection through work, is a multifocal concern [32]. This is a real concern for the assessment of the thermal stress associated with personal protective equipment among workers, too. Results of this survey confirmed a strong impact of COVID-19′s PPE in the heat stress perception of HCW, in line with the results obtained from similar studies carried out in England [27], in Germany [33], in Asia [6] but also in studies with wider participation [28]. The PCA identified 3 fundamental issues that represent the key elements on which to intervene in the management of the risk related to thermal stress in the health care sector: perception of heat stress conditions in the workplace and productivity loss;behavior during the working days;awareness of good practices.

Concerning the first issue, the fact that most of the workers (78.5%) declared to perceive heat stress conditions especially in the body areas covered by the PPE, confirms findings from a previous study carried out in England [27] that found a very similar percentage (72.3%). This aspect is certainly linked to the use of PPE for a high number of hours per day, as confirmed by Lee et al. [6] in two Asian countries, and which also determines important heat-related symptoms such as thirst, excessive sweating, fatigue, headache, difficulty concentrating, skin reactions and general discomfort conditions, with potential important effects on both the health and productivity of HCW. Some studies also described dark-colored urine, dizziness, muscle or abdominal cramps, gastrointestinal disturbance, rapid heartbeat [27] and mental symptoms [33] as phenomena associated with the use of PPE. Tabah et al. [28] showed that adverse effects of PPE (headaches, thirst and exhaustion) were associated with longer shift durations.

An important aspect to take into account, in relation to the results obtained from this and other studies investigating the heat stress perception in HCW, is that the work environments are generally conditioned. However, despite this aspect, workers still perceive thermal stress conditions which are therefore mainly caused by the intense workloads and the prolonged use of PPE declared by HCW. This is confirmed by the fact that workers who wear masks and gloves for a longer period of time (the most used devices for the HCW) are those who declared the worst thermal stress and general discomfort related to the use of COVID-19′s PPE. These aspects, related to the management of personnel engaged in the health emergency due to the pandemic, highlight the importance of adopting specific preventive customized strategies to protect workers, with information according to the task, the PPE worn and the work effort [9]. The importance of personalizing preventive strategies to safeguard the health and productivity of workers is one of the emerging priorities in the occupational field [28,34] and the underway pandemic only accentuates this need. A recent study [35] conducted by the pulmonology, intensive care and infectious diseases Hospital departments of two Italian cities, Bari and Foggia, on 116 healthcare workers directly involved in the healthcare of patients affected by COVID-19, underlined this need. In this study, each participant completed an online questionnaire aimed to investigate the impact of the COVID-19 pandemic on workers’ lifestyle changes and job performances. Comparing the results based on the type of mask (surgical mask vs. N95) used by each participant, the authors revealed that surgical masks reported a statistically higher average score for a greater number of disorders. In addition, considering the fact that this device is also used in the summer and outdoors by the general population, they suggested the importance of setting up a specific heat health warning system. Latest studies [36,37,38] highlighted that additional researches and comparative studies on various types of PPE are needed to determine optimal PPE for HCWs. In particular, their applicability in different environmental scenarios and in different situations of use must be tested. In a recent study [38], nineteen volunteers tested allocated head- or full body-ventilated PPE suits equipped with powered-air-purifying-respirators. This equipment was performed for different tasks during 6 working hours at 22 °C on one day and during 4 working hours at 28 °C on another day. Fluid loss, body temperature, heart rate was determined. Impaired visibility by flexible face shields, back pain related to the respirator of the fully ventilated suit and reduced dexterity due to multiple glove layers were major obstacles for workers. Heat stress and liquid loss were perceived as restrictive 28 °C but not 22 °C. These kinds of studies aimed at evaluating the duration and type of use of the main COVID-19-PPE are and will be increasingly fundamental in the perspective of the COVID-19 and other pandemic management.

The second issue “behavior during the working days” confirms this need because individual factors, such as work effort, tasks and the PPE, significantly influenced the negative behavior of workers during work shifts, such as refusing breaks to hydrate or rest because of overwork or fear of getting infected or to avoid taking off and re-wearing PPE. In particular, masks and gloves, especially if used for more than 4 h, were the PPE most related to negative behavior during the work shift. This finding highlights the importance of specific heat-related response plans ad-dressed to HCW with the aim of improve knowledge and promoting behavioral change to reduce thermal stress among workers. 

The awareness related to the importance of good practices to reduce heat stress risk appeared greater in workers who perceived warmer in the areas covered by PPE, with particular reference to maintaining a good level of hydration and keeping fit. These finding partially confirmed previous studies. Lee et al. [6] reported that although HCWs agreed that both hydration and aerobic fitness would increase heat tolerance, more workers perceived hydration as a better strategy than keeping fit. On the other hand, a recent meta-analysis showed that the most effective heat mitigation strategy was improving aerobic fitness, with hydration being least effective [39]. The effect of the aerobic fitness to reduce core temperature was shown in a study compared thermoregulatory and cardiovascular responses to heat stress before and after 8 weeks of endurance training in previously sedentary males [40] and in a subsequent study conducted by Mora-Rodriguez et al. [41] on endurance-trained and untrained cyclists. Furthermore, fitness can also enhance heat dissipation mechanisms [42], which is especially important when wearing PPE.

It is also interesting to observe how workers who used fewer types of masks and for a shorter period of time declared lower awareness of the importance of maintaining a good level of hydration and keeping fit. A different use of PPE could also explain the difference between doctors and nurses in the awareness of the importance of hydration. Another interesting aspect already observed in previous studies [43] is the use of crushed ice during work shifts to mitigate thermal stress and which has shown significance effects above all in relation to the use of headgear and the presence of a specific company procedure to dress PPE. In particular, Lee at al. [6] provided and administered to Singapore HCW an ice slurry made from a commercially available sports drink and a judgment on thermal comfort was requested before and after the ingestion with a scale from cold (+3) to hot (+3). The median rating improved from 2 (warm) before ingestion to 0 (neutral) after ingestion and so the authors concluded that the dual role of ice slurry to cool and hydrate HCW rendered it more beneficial than hydration with fluids and so this practice should be considered more often and also recommended. In fact, the effectiveness of ingesting ice slurry in the mitigation of heat stress and therefore in improving performance is also known in outdoor sports [44].

It is also important to consider that, although workers prevalently carry out their tasks in a conditioned environment, the summer period is still a critical period because workers may be exposed to heat stress conditions when they are out from work, for example during night rest [45,46]. This situation makes the worker more vulnerable as they are exposed to dehydration conditions away from working hours which represent a further critical factor that adds to the stress associated with the necessary use of PPE. A recent study [47] revealed that about 70% of workers initiate work with a suboptimal hydration status, meaning that workers are dehydrated at onset of work and that rehydration from day to day may be a bigger issue than failure to drink during the working shift.

The main strength of this study is that the results are suitable to be used in the operational field suggesting the creation of organizational solutions. These solutions can contribute to reduce the heat risk for HCWs, such as the creation of specific and personalized heat warning systems, supported by local real-time micrometeorological monitoring positioned in strategic hospital locations for the emergency management, the programming of work activities and the reorganization of spaces, as for example, the creation of dedicated rest areas where workers can safely remove their PPE without risking to get infected. This could allow not only to safeguard the health of workers but also their productivity and therefore ensure better management of the hospital emergency connected to the pandemic.

The main limitation of this study is represented by the small and unbalanced sample of HCWs, which is composed by mainly doctors or nurses and therefore it would be appropriate to extend the sample to other healthcare professions. A potential bias of our study, due to the absence of a sample plan strategy (planned as a second step of the study) for submitting our survey, has to be considered. In addition, the mode of self-administration online can be considered as a limit because the worker may have difficulties in understanding the items or devote little attention to the answers; while on the other hand, however, online administration can allow to reach a greater number of workers and can avoid the conditioning effect due to direct administration by an operator too. Another limitation of the research is represented by the lack of simultaneous continuous microclimatic monitoring in the workplace and this aspect will have to be taken into consideration in subsequent studies in order to quantify the real thermal environment and its influence on the HCW heat stress perception. 

The survey will be replicated during the summer of 2021 to increase the sample size with particular reference to the involvement of different categories of healthcare professions. Furthermore, by exploiting the results obtained with this first study, and especially the PCA, the questionnaire will be simplified. The simplification of the questionnaire will make it easier and faster to compile and hopefully workers will be more enticed to participate in the survey.

## 5. Conclusions

The COVID-19 pandemic emergency combined with workload for healthcare professionals call for the further implementation of adaptation strategies and specific interventions to respond to thermal stress of health and social care staff; thus, preserving both workers’ health and productivity, with positive effects on the management of the health emergency linked to the pandemic. 

Our findings are important for promoting and suggesting prevention measures in order to identify organizational and procedural solutions to reduce thermal stress for HCWs engaged in the management of the COVID-19 emergency, and also for potential future similar emergencies. In fact, the reorganization of internal hospital spaces, the creation of safe rest areas, where it is possible to respect the safety distances and temporarily take off the PPE, do not represent very complex and expensive solutions to be implemented. These relatively simple solutions can be a great help to safeguard the HCW. Imposing mandatory breaks in case of high environmental temperatures, or strict enforcement of specific work/rest ratios to limit the duration of PPE use, should also be considered. In addition, the adoption of company procedures designed to guide the worker to dress and remove the PPE with areas dedicated to this purpose could have a positive impact on the management of the emergency. The study reports a high perception of thermal stress among HCWs despite the fact that work environments are prevalently indoor and air conditioned, demonstrating the importance of individual factors such as workload and the type of clothing worn (PPE) in heat stress perception. This suggests the importance of adopting preventive heat-related strategies also including the personalization of information by developing appropriate heat health warning systems addressed to the occupational sector. A microclimatic monitoring in some strategic hospital areas should be considered too, in order to provide real-time information and therefore facilitating the emergency management plan. It would be desirable to implement national programs for the safeguard of HCW from heat stress, in line with national occupational health and safety policies. In conclusion, even in the health care sector, that might seem “more protected” from the effects of heat—because mainly indoors and in air-conditioned environments—the development of standards, guidelines, and codes of practice represent a priority in order to protect often vulnerable workers due to the prolonged use of PPE and the exposure times caused by COVID-19 emergency.

## Figures and Tables

**Figure 1 ijerph-18-03861-f001:**
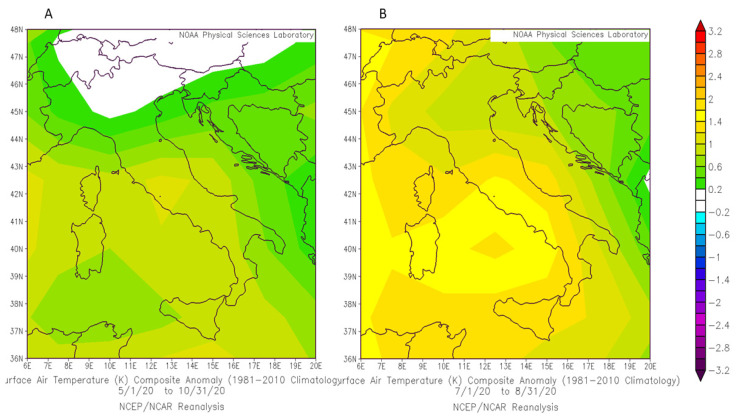
Air temperatures anomalies in Italy during the period May–October 2020 (**A**) and during the period July-August 2020 (**B**) compared to the climatology 1981–2010. Data obtained from https://psl.noaa.gov/cgi-bin/data/composites/printpage.pl, accessed on 6 April 2021.

**Figure 2 ijerph-18-03861-f002:**
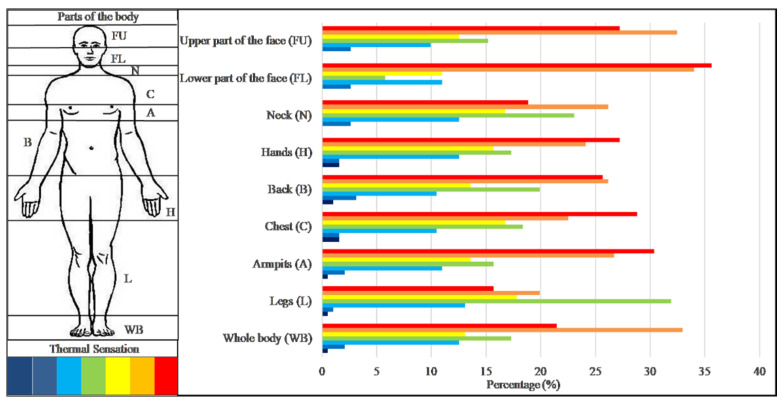
Thermal sensations declared by healthcare workers for each parts of the body covered by PPE during working time. Dark blue: Very cold; Blue: Cold; Light blue: Slightly cold; Green: Neutral; Yellow: Slightly hot; Orange: Hot; Red: Very hot.

**Figure 3 ijerph-18-03861-f003:**
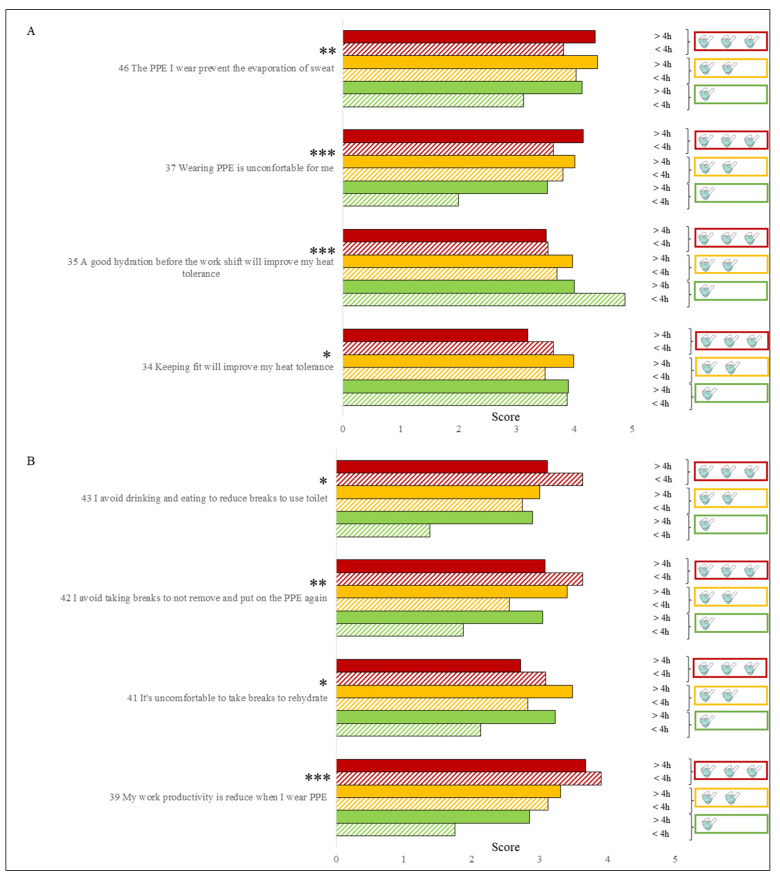
ANOVA to evaluate the effect of the use of the masks on the answers related to items 34, 35, 37 and 46 (**A**) and items 39, 41, 42 and 43 (**B**). Different kind of masks (from 1 up to 3 kind) and different time of use (<4 h or >4 h) was considered. A 5-point Likert scale (1 for strongly disagree and 5 for strongly agree) was used for questions. (Sig): *** *p* < 0.001; ** *p* < 0.01; * *p* < 0.05.

**Figure 4 ijerph-18-03861-f004:**
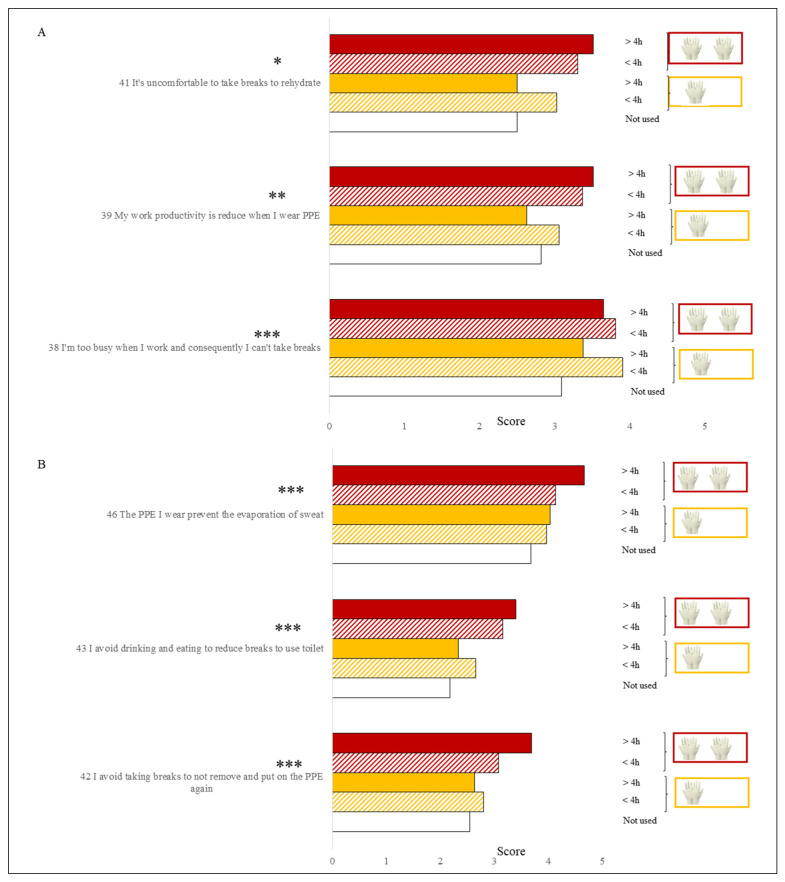
ANOVA to evaluate the effect of the use of the gloves on the answers related to items 38, 39, 41 (**A**) and items 42, 43, 46 (**B**). One or two pairs of overlapping gloves and dif-ferent time of use (<4 h or >4 h) was considered. A 5-point Likert scale (1 for strongly disagree and 5 for strongly agree) was used for questions. (Sig): *** *p* < 0.001; ** *p* < 0.01; * *p* < 0.05.

**Table 1 ijerph-18-03861-t001:** Results of the questionnaire submitted to healthcare workers.

Healthcare Workers		N ^1^	% ^2^
Do you avoid eating on fast days for personal reasons?	Never	167	87.4
	Sometimes	7	3.7
	Often	14	7.3
	Very often	1	0.5
	Ever	2	1.0
How do you judge your work effort on average?	At rest	2	1.0
	Lightweight	13	6.8
	Moderate	65	34.0
	High	82	42.9
	Very high	29	15.2
How do you judge the thermal environment in which you generally work?	Very cold	6	3.1
	Cold	11	5.8
	Slightly cold	20	10.5
	Neutral	40	20.9
	Slightly hot	33	17.3
	Hot	53	27.7
	Very hot	28	14.7
For how many hours do you usually wear N95 mask or equivalent (FFP2)?	0 h	39	20.4
	1 to 3 h	50	26.2
	4 to 6 h	65	34.0
	over 6 h	37	19.4
For how many hours do you usually wear FFP3 mask?	0 h	146	76.4
	1 to 3 h	28	14.7
	4 to 6 h	8	4.2
	over 6 h	9	4.7
How many hours do you usually wear a surgical mask?	0 h	17	8.9
	1 to 3 h	32	16.8
	4 to 6 h	69	36.1
	over 6 h	73	38.2
How many hours do you usually wear gloves (one pair)?	0 h	40	20.9
	1 to 3 h	66	34.6
	4 to 6 h	55	28.8
	over 6 h	30	15.7
How many hours do you usually wear gloves (two pairs)?	0 h	81	42.4
	1 to 3 h	62	32.5
	4 to 6 h	29	15.2
	over 6 h	19	9.9
How many hours do you usually wear a disposable gown?	0 h	48	25.1
	1 to 3 h	71	37.2
	4 to 6 h	50	26.2
	over 6 h	22	11.5
How many hours do you usually wear a normal gown?	0 h	94	49.2
	1 to 3 h	27	14.1
	4 to 6 h	43	22.5
	over 6 h	27	14.1
How many hours do you usually wear a disposable apron?	0 h	155	81.2
	1 to 3 h	21	11.0
	4 to 6 h	11	5.8
	over 6 h	4	2.1
How many hours do you usually wear disposable glasses?	0 h	77	40.3
	1 to 3 h	41	21.5
	4 to 6 h	50	26.2
	over 6 h	23	12.0
How many hours do you usually wear a disposable visor?	0 h	78	40.8
	1 to 3 h	56	29.3
	4 to 6 h	38	19.9
	over 6 h	19	9.9
How many hours do you usually wear disposable headgear?	0 h	67	35.1
	1 to 3 h	34	17.8
	4 to 6 h	49	25.7
	over 6 h	41	21.5
How many hours do you usually wear disposable closed boots or work shoes?	0 h	119	62.3
	1 to 3 h	13	6.8
	4 to 6 h	21	11.0
	over 6 h	38	19.9
How many hours do you usually wear shoes covers?	0 h	102	53.4
	1 to 3 h	44	23.0
	4 to 6 h	32	16.8
	over 6 h	13	6.8
How many hours do you usually wear sanitary clogs?	0 h	73	38.2
	1 to 3 h	5	2.6
	4 to 6 h	42	22.0
	over 6 h	71	37.2
How many days per week do you use PPE at work?		5.2	SD 1.0
How long (minutes) does it take you to wear PPE at the start of the work shift?		7.1	SD 5.5
Do you work mainly in an air-conditioned environment?	Yes	151	79.1
	No	40	20.9
Is there a company procedure that allows you to remove PPE during work breaks?	Yes	106	55.5
	No	85	44.5
If yes, when? More than one answer is possible	In the middle of the day	39	20.4
	When i go to the toilet	31	16.2
	After each visit	31	16.2
	Whenever i need to	42	22.0
Is there a dedicated rest area in your workplace?	Yes	88	46.1
	No	103	53.9
How do you try and reduce heat stress when using PPE? It is possible to select more than one answer for this question.	I often drink water	108	56.5
	I drink ice cold drinks	1	0.5
	I take breaks whenever possible	81	42.4
	I try to dress in light clothing	90	47.1
	Breathing techniques	28	14.7
	I prefer ventilated and cool environments	64	33.5
Heat stress in the areas covered by the PPE		150	78.5
Symptoms generally perceived when I wear PPE	Thirst	111	58.1
	Excessive sweating	135	70.7
	Fatigue	88	46.1
	Headache	82	42.9
	Difficulty concentrating	56	29.3
	Skin reaction	51	26.7
	General discomfort	99	51.8
What is your thermal sensation when you wear PPE during work activities?	Neutral	2	1.0
	Slightly hot	21	11.0
	Hot	68	35.6
	Very hot	100	52.4
Productivity loss perception caused by heat stress		155	81

^1^ N, sample size; ^2^ % percentage of the sample.

**Table 2 ijerph-18-03861-t002:** Principal Component Analysis of section “Worker’s knowledge about thermal stress and attitudes towards PPE use”. Extraction method: Principal component analysis. Rotation method: Varimax with Kaiser normalization.

N-Item	Component		
	1 “Perception of Heat Stress Conditions in the Workplace and of Productivity Loss”	2 “HCW Behavior during the Working Days”	3 “Awareness of Good Practices”
29-Heat stress can impair my reasoning	0.873		
31-Heat Stress can affect my psychological state	0.829		
33-Heat stress can negatively affect my commitment at work	0.813		
28-Heat stress can affect my productivity	0.790		
32-Heat stress can negatively affect my emoticons	0.788		
30-Heat Stress can affect my physical health	0.765		
42-I avoid taking breaks to not remove and put on the PPE again		0.863	
43-I avoid drinking and eating to reduce breaks to use toilet		0.849	
41-It is uncomfortable to take breaks to rehydrate		0.763	
44-I avoid taking breaks to reduce the risk of getting infected		0.715	
35-A good hydration before the work shift will improve my heat tolerance			0.875
34-Keeping fit will improve my heat tolerance			0.824
36-Adequate rest between shifts will improve my tolerance			0.746

**Table 3 ijerph-18-03861-t003:** Difference between groups concerning issues related to the effects of the heat stress on workers and their productivity loss perception.

	Item		Kind of Work		Thermal Sensation with PPE		Glasses and Visor		Headgear	
N°		G	M(SD)	F/Sig	M(SD)	F/Sig	M(SD)	F/Sig	M(SD)	F/Sig
29	Heat stress can impair my reasoning	1	4.2 (0.9)	ns	4.0 (1.1)	ns	4.0 (1.1)	ns	**3.9 (1.0)**	**4.2 ***
		2	4.2 (1.1)		4.0 (0.9)		4.2 (1.0)		**4.1 (1.0)**	
		3	4.2 (1.0)		4.3 (1.0)		4.3 (1.0)		**4.4 (1.0)**	
30	Heat Stress can affect my physical health	1	4.0 (1.0)	ns	**4.0 (1.2)**	**4.4 ****	3.9 (1.2)	ns	**3.9 (1.2)**	**6.1 ****
		2	4.3 (0,9)		**4.0 (1.1)**		4.2 (0.9)		**4.2 (0.9)**	
		3	4.2 (0.9)		**4.4 (0.8)**		4.3 (0,8)		**4.4 (0.8)**	
31	Heat Stress can affect my psychological state	1	4.2 (0.9)	ns	4.1 (1.0)	ns	4.3 (0.9)	ns	4.2 (1.0)	Ns
		2	4.3 (0.9)		4.2 (1.0)		3.3 (0.9)		4.1 (1.0)	
		3	4.2 (0.9)		4.4 (0.9)		4.3 (0.9)		4.4 (0.8)	
32	Heat stress can negatively affect my emoticons	1	4.1 (1.1)	ns	4.0 (1.3)	ns	4.1 (1.1)	ns	4.0 (1.2)	Ns
		2	4.2 (1.0)		4.0 (1.1)		4.2 (1.0)		4.2 (1.1)	
		3	4.0 (1.2)		4.2 (1.1)		4.0 (1.2)		4.2 (1.1)	
33	Heat stress can negatively affect my commitment at work	1	4.1 (1.1)	ns	3.9 (1.4)	ns	4.0 (1.1)	ns	3.9 (1.1)	Ns
		2	4.1 (1.2)		4.0 (1.1)		4.2 (1.2)		4.1 (1.3)	
		3	4.1 (1.1)		4.2 (1.1)		4.1 (1.2)		4.3 (1.1)	
46	The PPE I wear prevent the evaporation of sweat	1	4.2(1.0)	ns	**3.5 (1.1)**	**7.1 *****	**3.9 (1.0)**	**6.9 *****	3.8 (1.1)	Ns
		2	4.3 (1.0)		**4.1 (0.8)**		**3.9 (1.2)**		4.0 (1.0)	
		3	4.0(1.1)		**4.4 (1.1)**		**4.5 (0.9)**		4.5 (0.9)	
37	Wearing PPE is uncomfortable for me	1	**4.1 (1.0)**	**5.7 ****	**2.9 (1.4)**	**10.6 *****	**3.7 (1.2)**	ns	3.7 (1.2)	Ns
		2	**3.7 (1.2)**		**3.6 (1.1)**		**3.6 (1.4)**		3.6 (1.3)	
		3	**3.4 (1.3)**		**4.1 (1.1)**		**4.0 (1.1)**		3.9 (1.1)	
39	My work productivity is reduced when I wear PPE	1	3.2 (1.2)	ns	**2.6 (1.3)**	**7.7 *****	3.0 (1.0)	ns	3.1 (1.2)	Ns
		2	3.2 (1.2)		**2.9 (1.1)**		3.2 (1.4)		3.2 (1.2)	
		3	3.1 (1.2)		**3.5 (1.2)**		3.3 (1.3)		3.3 (1.2)	
28	Heat stress can affect my productivity	1	4.3 (0.8)	ns	**4.1 (1.1)**	**5.3 ****	4.4 (1.0)	ns	**4.0 (1.0)**	**4.6 ****
		2	4.4 (0.9)		**4.1 (0.9)**		4.3 (0.8)		**4.3 (0.8)**	
		3	4.2 (1.0)		**4.5 (0.8)**		4.4 (0.8)		**4.5 (0.9)**	

Group (G): Kind of work (1 general practitioner and hospital doctor, 2 nurse/pediatric nurse, 3 other); Thermal sensation with PPE (1 neutral or slightly hot, 2 hot, 3 very hot); Glasses and visor (1 not used, 2 from one to three hours, 3 more than four hours); Headgear (1 not used, 2 from 1 h to four hours, 3 more than four hours). A 5-point Likert scale (1 for strongly disagree and 5 for strongly agree) was used for questions. M is the Mean value; F is Fisher–Snedecor distribution; in brackets Standard deviation (SD). (Sig): *** *p* < 0.001; ** *p* < 0.01; * *p* < 0.05 and values in bold.

**Table 4 ijerph-18-03861-t004:** Difference between groups concerning issues related to the effects of the heat stress on workers and their productivity loss perception.

	Item		Work Effort		Company Procedure to Dress PPE		Rest Area		Parts of the Body	
N°		G	M(SD)	F/Sig	M(SD)	F/Sig	M(SD)	F/Sig	M(SD)	F/Sig
29	Heat stress can impair my reasoning	1	4.1 (1.0)	ns	**4.0 (1.1)**	**4.8 ***	4.2 (1.0)	ns	4.2 (1.0)	ns
		2	4.2 (1.0)		**4.4 (0.8)**		4.1 (1.0)		4.3 (1.0)	
30	Heat Stress can affect my physical health	1	4.0 (1.0)	ns	4.1 (1.0)	ns	4.2 (0.9)	ns	4.2 (0.9)	ns
		2	4.3 (0.9)		4.3 (0.9)		4.2 (1.0)		4.1 (1.1)	
31	Heat Stress can affect my psychological state	1	4.2 (1.0)	ns	**4.1 (1.0)**	**6.1 ****	4.3 (0.9)	ns	4.2 (1.0)	ns
		2	4.3 (0.9)		**4.5 (0.8)**		4.2 (1.0)		4.4 (0.7)	
32	Heat stress can negatively affect my emoticons	1	4.0 (1.2)	ns	3.9 (1.2)	ns	4.0 (1.1)	ns	4.1 (1.1)	ns
		2	4.1 (1.1)		4.3 (0.9)		4.2 (1.1)		3.9 (1.2)	
33	Heat stress can negatively affect my commitment at work	1	4.2 (1.1)	ns	**3.9 (1.2)**	**4.4 ***	4.1 (1.2)	ns	4.1 (1.2)	ns
		2	4.0 (1.2)		**4.3 (1.0)**		4.1 (1.1)		4.1 (1.2)	
46	The PPE I wear prevent the evaporation of sweat	1	4.1 (1.0)	ns	4.1 (1.1)	ns	4.2 (1.0)	ns	4.2 (1.1)	ns
		2	4.2 (1.0)		4.3 (0.9)		4.1 (1.1)		4.2 (0.9)	
37	Wearing PPE is unconfortable for me	1	**3.6 (1.3)**	**4.4 ***	3.5 (1.3)	ns	3.7 (1.1)	ns	3.7 (1.2)	ns
		2	**3.9 (1.1)**		4.1 (1.0)		3.8 (1.3)		3.9 (1.2)	
39	My work productivity is reduced when I wear PPE	1	3.0 (1.2)	ns	**3.0 (1.2)**	**4.9 ***	3.2 (1.3)	ns	**3.1 (1.2)**	**6.2 ****
		2	3.3 (1.3)		**3.4 (1.2)**		3.2 (1.2)		**3.6 (1.2)**	
28	Heat stress can affect my productivity	1	4.2 (0.8)	ns	**4.1 (1.0)**	**7.9 *****	4.3 (0.9)	ns	4.3 (0.9)	ns
		2	4.3 (0.9)		**4.5 (0.7)**		4.3 (0.9)		4.3 (1.0)	

Group (G): Work effort (1 from moderate to rest, 2 from high to very high); Company procedure to dress PPE (1 yes, 2 no); Rest area (1 yes, 2 no); Different perception between parts of the body covered by PPE (1 yes, 2 no). A 5-point Likert scale (1 for strongly disagree and 5 for strongly agree) was used for questions. M is the Mean value; F is Fisher–Snedecor distribution; in brackets Standard deviation (SD). (Sig): *** *p* < 0.001; ** *p* < 0.01; * *p* < 0.05 and values in bold.

**Table 5 ijerph-18-03861-t005:** Difference between groups concerning issues related to worker’s behavior and awareness of good practices to increase the heat tolerance.

	Item		Kind of Work		Thermal Sensation with PPE		Glasses and Visor		Headgear	
N°		G	M (SD)	F/Sig	M(SD)	F/Sig	M(SD)	F/Sig	M(SD)	F/Sig
38	I’m too busy when I work and consequently I can’t take breaks	1	3.7 (1.1)	ns	3.5 (1.1)	ns	3.6 (1.1)	ns	3.5 (1.1)	ns
		2	3.7 (1.1)		3.7 (1.1)		3.6 (1.2)		3.8 (1.0)	
		3	3.5 (1.2)		3.6 (1.1)		3.7 (1.1)		3.7 (1.1)	
41	It is uncomfortable to take breaks to rehydrate	1	3.1 (1.4)	ns	2.8 (1.2)	ns	**2.9 (1.1)**	**5.7 ****	**2.8 (1.2)**	**8.0 *****
		2	3.4 (1.4)		3.1 (1.3)		**2.7 (1.4)**		**2.6 (1.5)**	
		3	2.8 (1.4)		3.2 (1.4)		**3.5 (1.4)**		**3.5 (1.4)**	
42	I avoid taking breaks to not remove and put on the PPE again	1	3.2 (1.4)	ns	2.4 (1.3)	ns	**2.7 (1.2)**	**5.5 ****	**2.7 (1.4)**	**6.5 *****
		2	3.2 (1.5)		3.0 (1.3)		**2.8 (1.5)**		**2.6 (1.4)**	
		3	2.7 (1.5)		3.2 (1.5)		**3.4 (1.5)**		**3.5 (1.5)**	
43	I avoid drinking and eating to reduce breaks to use toilet	1	**3.1 (1.5)**	**7.9 *****	2.6 (1.6)	ns	2.6 (1.5)	ns	**2.6 (1.5)**	**3.3 ***
		2	**3.2 (1.4)**		2.9 (1.4)		2.8 (1.5)		**2.9 (1.6)**	
		3	**2.2 (1.3)**		3.0 (1.5)		3.2 (1.4)		**3.2 (1.4)**	
44	I avoid taking breaks to reduce the risk of getting infected	1	2.7 (1.4)	ns	**2.9 (1.5)**	**3.1 ***	2.5 (1.4)	ns	2.5 (1.4)	ns
		2	2.9 (1.5)		**2.9 (1.4)**		3.0 (1.4)		3.0 (1.4)	
		3	2.7 (1.5)		**2.7 (1.5)**		2.8 (1.5)		2.9 (1.5)	
40	It is important to keep hydrated during the work shift	1	4.4 (0.9)	ns	4.6 (0.7)	ns	4.4 (0.7)	ns	4.5 (0.7)	ns
		2	4.5 (0.8)		4.3 (0.9)		4.6 (0.6)		4.4 (0.8)	
		3	4.6 (0.7)		4.6 (0.8)		4.6 (0.8)		4.5 (0.9)	
45	Slush drinks improve my tolerance to heat	1	2.1 (1.2)	ns	2.0 (1.5)	ns	2.2 (1.2)	ns	**2.3 (1.3)**	9.8 ***
		2	2.1 (1.3)		2.0 (1.1)		2.2 (1.3)		**2.1 (1.1)**	
		3	2.0 (1.2)		2.1 (1.2)		1.8 (1.1)		**1.8 (1.1)**	
34	Keeping fit will improve my heat tolerance	1	3.9 (1.1)		**4.3 (0.8)**	**4.7 ****	3.9 (1.1)	ns	3.7 (1.1)	ns
		2	3.6 (1.2)	ns	**3.9 (1.1)**		3.7 (1.1)		3.5 (1.2)	
		3	3.7 (1.2)		**3.5 (1.2)**		3.6 (1.2)		3.8 (1.2)	
35	A good hydration before the work shift will improve my heat tolerance	1	3.9 (1.1)	**3.4 ***	**4.6 (0.7)**	**6.6 *****	4.0 (1.1)	ns	4.0 (1.0)	ns
		2	3.6 (1.1)		**3.9 (1.2)**		4.0 (1.1)		3.7 (1.2)	
		3	4.2 (1.0)		**3.7 (1.1)**		3.8 (1.1)		3.9 (1.1)	
36	Adequate rest between shifts will improve my tolerance	1	**4.3 (1.0)**	**3.3 ***	4.4 (0.9)	ns	4.2 (0.9)	ns	4.3 (0.9)	ns
		2	**3.9 (1.2)**		4.3 (0.9)		4.1 (1.2)		3.9 (1.3)	
		3	**4.4 (0.9)**		4.0 (1.2)		4.2 (1.1)		4.2 (1.0)	

Group (G): Kind of work (1 general practitioner and hospital doctor, 2 nurse/pediatric nurse, 3 other); Thermal sensation with PPE (1 neutral or slightly hot, 2 hot, 3 very hot); Glasses and visor (1 not used, 2 from one to three hours, 3 more than four hours); Headgear (1 not used, 2 from 1 h to four hours, 3 more than four hours). A 5-point Likert scale (1 for strongly disagree and 5 for strongly agree) was used for questions. M is the Mean value; F is Fisher–Snedecor distribution; in brackets Standard deviation (SD). (Sig): *** *p* < 0.001; ** *p* < 0.01; * *p* < 0.05 and values in bold.

**Table 6 ijerph-18-03861-t006:** Difference between groups concerning issues related to worker’s behavior and awareness of good practices to increase the heat tolerance.

	Item		Work Effort	Company Procedure to Dress PPE			Rest Area		Parts of the Body	
N°		G	M(SD)	F/Sig	M(SD)	F/Sig	M(SD)	F/Sig	M(SD)	F/Sig
38	I’m too busy when I work and consequently I can’t take breaks	**1**	**3.5 (1.1)**	**4.5 ***	3.5 (1.1)	ns	**3.5 (1.1)**	**4.5 ***	**3.6 (1.1)**	**4.6 ***
		**2**	**3.8 (1.1)**		3.8 (1.1)		**3.8 (1.1)**		**4.0 (1.0)**	
41	It’s uncomfortable to take breaks to rehydrate	**1**	**2.8 (1.3)**	**5.6 ****	3.1 (1.3)	ns	**2.9 (1.3)**	**4.1 ***	**2.9 (1.4)**	**13.8 *****
		**2**	**3.3 (1.4)**		3.3 (1.4)		**3.3 (1.5)**		**3.8 (1.2)**	
42	I avoid taking breaks to not remove and put on the PPE again	1	2.8 (1.5)	ns	2.9 (1.4)	ns	3.0 (1.4)	ns	**2.9 (1.4)**	**5.0 ***
		2	3.2 (1.4)		3.3 (1.5)		3.1 (1.5)		**3.5 (1.5)**	
43	I avoid drinking and eating to reduce breaks to use toilet	1	2.8 (1.5)	ns	2.8 (1.4)	ns	2.9 (1.5)	ns	2.8 (1.4)	ns
		2	3.0 (1.5)		3.1 (1.5)		2.9 (1.5)		3.2 (1.6)	
44	I avoid taking breaks to reduce the risk of getting infected	1	2.7 (1.4)	ns	2.6 (1.4)	ns	2.8 (1.4)	ns	2.7 (1.4)	ns
		2	2.8 (1.5)		2.9 (1.4)		2.7 (1.5)		3.1 (1.5)	
40	It is important to keep hydrated during the work shift	1	4.4 (0.9)	ns	4.4 (0.9)	ns	4.5 (0.7)	ns	4.5 (0.8)	ns
		2	4.6 (0.8)		4.6 (0.6)		4.5 (0.9)		4.4 (0.8)	
45	Slush drinks improve my tolerance to heat	1	2.1 (1.1)	ns	**1.8 (1.1)**	**7.3 *****	2.2 (1.3)	ns	2.1 (1.2)	ns
		2	2.0 (1.3)		**2.3 (1.2)**		1.9 (1.1)		1.9 (1.2)	
34	Keeping fit will improve my heat tolerance	1	3.6 (1.1)	ns	**3.9 (1.1)**	**5.3 ***	3.7 (1.2)	ns	3.8 (1.1)	ns
		2	3.8 (1.2)		**3.5 (1.2)**		3.8 (1.1)		3.6 (1.1)	
35	A good hydration before the work shift will improve my heat tolerance	1	3.7 (1.2)	ns	4.0 (1.1)	ns	3.8 (1.1)	ns	3.9 (1.1)	ns
		2	4.0 (1.0)		3.7 (1.1)		3.9 (1.1)		3.7 (1.2)	
36	Adequate rest between shifts will improve my tolerance	1	**4.0 (1.2)**	**5.2 ***	4.3 (0.9)	Ns	**4.0 (1.1)**	**4.7 ***	4.2 (1.0)	ns
		2	**4.3 (0.9)**		4.0 (1.2)		**4.3 (1.0)**		3.9 (1.2)	

Group (G): Work effort (1 from moderate to rest, 2 from high to very high); Company procedure to dress PPE (1 yes, 2 no); Rest area (1 yes, 2 no); Different perception between parts of the body covered by PPE (1 yes, 2 no). A 5-point Likert scale (1 for strongly disagree and 5 for strongly agree) was used for questions. M is the Mean value; F is Fisher–Snedecor distribution; in brackets Standard deviation (SD). (Sig): *** *p* < 0.001; ** *p* < 0.01; * *p* < 0.05 and values in bold.

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
