# Peer review of "A Web Survey to Evaluate the Thermal Stress Associated with Personal Protective Equipment among Healthcare Workers during the COVID-19 Pandemic in Italy†"

_ijerph, 2021, doi:10.3390/ijerph18083861_

Round 1

Reviewer 1 Report

Covid-Heat Review

While in general this is a sound article I think it goes a step too far in linking these effects to heat waves and indeed to high ambient temperatures.  I could not see any direct evidence to justify this and so references to external heat (there are not very many) should either be justified/quantified or they should be removed from the article.  See more detail below.  While I recommend a revision, it is not a large revision.

Line46 “exponential” increase.   Exponential is a specific mathematical function – do you really mean exponential?  If so, please give an example.

Line 49 “during the warm season” – I fail to see how the season can affect the outcome when air-conditioning would have easily coped with an anomaly in temperature of a maximum of 1.5C (line 155).

One of the questions in the table says “Do you work mostly in an air conditioned environment?”

Don’t you think it would be prudent to separate out those responses in an “air conditioned environment” and distinguish that from non-air conditioned environments.  After all the paper is about working in hot environments – it talks about “over-heating …. in hot environments”, “heat health warning systems”, “over heating amplifying the heating in hot weather” etc etc.  Yet over 50% of workers say they are NOT working in either hot or very hot environments and indeed some are working in cold environments.  Were your results broken down to those working in hot environments vs those working in non-hot environments.  If the results of these two groups are combined then the results while still pertaining to PPE gear will not pertain to hot environments.

It is very surprising that “did not show any significant difference between working environments with or without air conditioning.”  Also results “did not show significant differences between workplaces belonging to different geographical areas” – and elsewhere in the document it was stated that southern regions were hotter.   This of course means that it is not the hot environment that is the problem but the clothing and the work.  So all the sentences applying to “hot environments”, “hot weather”, “heat warning systems” etc do not apply.

Line 474 “fitness can also enhance heat dissipation mechanisms [36], which is especially important when wearing PPE.”  While I understand this to be true in a normal “sporting” environment with minimal clothes, I don’t understand how this would work with PPE gear that prevents equal heat loss from both fit and unfit people.

Line 532 “pandemic emergency combined with extreme heat wave events” – I don’t think the results can be used to reach this conclusion (as discussed above).

I suggest someone carefully proof-read the document for grammatical errors.  Here are a few I picked up on the way – this list is by no means exhaustive:

Line 52-58 is a very long sentence making it hard to read.

Line 61 – at least change “worn” to “wear”.  Other changes could be made for that sentence to read better.

Line 62  I don’t understand why protection against nuclear agents, chemical … and radiological risks” is relevant to this discussion.

Line 465 “feet” should be replace by “fit”.

Reviewer 2 Report

The work proposal entitled "A web survey to evaluate the thermal stress associated with personal protective equipment among healthcare workers during the COVID-19 pandemic in Italy" is interesting and has a relevant scientific and biometeorological contribution to the current pandemic scenario. However, there is a need to expand the discussions of the results in the face of a greater number of bibliographic references and hard-hitting studies that border the research theme.

The authors shyly explore the vast bibliography on the impact of clothing on comfort and thermal sensation.

There is little emphasis on the concrete impacts of this research for health professional users.

Reviewer 3 Report

The study focuses on the increase of the thermal discomfort experimented by healthcare workers (doctors and nurses in particular) during the pandemic. The study in its totality is interesting and the paper is pleasant to read. Beyond the contribution to the thermal comfort research, this work has a great social value, in particular in denouncing the consequences (often forgotten) of the incredible rhythm of work that healthcare professionals are sustaining since the pandemic started.

Furthermore, the solutions proposed by the authors are extremely simple and concrete, so it would be easy to put them into practice.

In my opinion, the method and results are congruent, the survey is well designed and the comparison among groups is engaging. I especially appreciated the analysis of the thermal response of different metabolic conditions.

What I miss the most in the article, considering thermal comfort evaluation is the main topic, is some attempt at quantifying the described phenomena. For instance, how much the clothing level increase, approximately, due to the additional garments the workers have to wear? Or, how much the metabolic rate could decrease with the proposed solutions (let us say with the ice slurry)? This could be a suggestion for the upcoming new survey.

Please receive the following comments as suggestions to help the reader’s comprehension:

-Figures are mentioned in the text, but I could not find them in the article nor in the supplementary file.

-line 147-148. Is this specification here really necessary? I understand the specification later on, when talking about the average outdoor temperature, but not here. Last time I checked, North and Center-South of Italy still belong to the same country, so ‘[…] HCW who perform activities in Italian hospitals.’ would be more than enough to understand you are referring to the whole territory.

-line 154-155. It would be interesting to explain how you calculated thermal anomalies.

-line 179-180. I think political correctness should be a must in every scientific text, and maybe I am wrong, but I am reading between the lines an unnecessary reference to religion. Why, according to you, being Italian should be clearly explaining the first sentence? If I were you, I would add a clarification to the statement or, even better, I would delete the second part. Also, fasting could be related to phenomena that you are not considered: for instance, intermitted fasting is a practice quite in fashion all over the world.

-Table 1. Some questions are repeated twice.

-Table 2. More explanation is needed. Researchers that are not expert in statistics can get lost here.

-line 414-416. I would rephrase the sentence (or split in two, better): it took me three times to understand the meaning.

-line 465. Fit is misspelt.

-line 510-520. Personally, I would move this section to the introduction, as an explanation of why you chose the e-research. If you are discussing the limitations of the work, it is more congruent to leave in this section just the previous part (difficulties in understanding/ little attention to the answers).

Round 2

Reviewer 1 Report

There has been great improvement in the manuscript.  I am still not convinced that the environmental heat is an issue.  I think its the stress including heat stress in wearing the protective gear that is an issue.  For example:

"It is also important to consider that, although workers prevalently carry out their tasks in a conditioned environment, the summer period is still a critical period because workers may be exposed to heat stress conditions when they are out from work, for example during night rest [45, 46]. "

A quick check around Italy for maximum night time temperatures shows that during the night in summer the temperature is almost always below 22C.  This is more than adequate for night time cooling as most heat is generated while physically working and as the person is resting it does not make sense that there is inadequate cooling at night.

However the article is of sufficient interest and importance for me to leave this just as a comment and hopefully in the next round of research more care will be taken to separate out the effect of heat as distinct from the effect of discomfort in the workers wearing PPE gear for extended periods.